# Design, Fabrication, and Characterization of an Impact Dissipative Layer for Fiber-Reinforced Polymer Composites

**DOI:** 10.3390/polym14173631

**Published:** 2022-09-02

**Authors:** Cristina Alía-García, Álvaro Rodríguez-Ortiz, Patrick Townsend, Juan C. Suárez-Bermejo

**Affiliations:** 1ETS de Ingeniería y Diseño Industrial, Universidad Politécnica de Madrid, Rda. de Valencia, 3, 28012 Madrid, Spain; 2Escuela Superior Politécnica del Litoral (ESPOL), Facultad de Ingeniería Marítima y Ciencias del Mar (FIMCBOR), Guayaquil 09-01-5863, Ecuador; 3Departamento de Materiales, ETS de Ingenieros Navales, Universidad Politécnica de Madrid, Av., de la Memoria, 4, 28040 Madrid, Spain

**Keywords:** impact energy, GFRP laminates, auxetic, dissipative layer, damage

## Abstract

This study presents the option of an effective low-impact energy dissipating material applied to GFRP (glass fiber reinforced plastic) composite laminates using auxectic technology in the case of planing hull vessels that use the same high-speed light materials that repeatedly impact the surface of the water when sailing, producing a slamming phenomenon. Research shows that the option to modify the laminate with an auxectic layer protects the laminate from damage. This work proposes the manufacturing of dissipative layers, introduced in laminates made with a polymeric matrix and fiberglass reinforcement, which are evaluated with weight drop tests under different impact energies. The data are collected and processed by a unidirectional gravitometer that gives the acceleration values of the impactor. The tests compare unmodified panels with modified panels, showing that the energy absorbed by the unmodified panel is greater at equal energy levels. The returned energy comparison curve is shown, and the benefits of its use are presented.

## 1. Introduction

The availability of increasingly lighter and stronger materials with high energy dissipation capacity and damage tolerance is essential for many applications of industrial interest. When there are no weight restrictions, there are very effective materials capable of dissipating the mechanical energy coming from an impact. However, when the use of these materials is not possible due to density or thickness limitations, the need arises to obtain new structural hybrid materials that meet these requirements.

The marine sector and, specifically, high-speed boats are examples of the need to develop new, lighter materials without reducing their mechanical properties. Furthermore, for many years, these vessels were designed to perform well in calm waters. However, sea conditions, together with technological advances that allow for faster speeds in the vessels, cause the occurrence of phenomenon that compromises the integrity of vessel materials [1]. The present study aims to investigate this phenomenon because planing hull vessels are the most common ships for maritime use. One of these phenomena is slamming, which occurs when the bow of the vessel rises above the water surface, producing a sudden change in the acceleration of the ship and falling again when the hull collides with this fluid-free surface. This phenomenon produces, in the hull of high-speed boats, elevated pressure pulses of short duration (on the order of milliseconds), localized on a very small surface as analyzed by researchers [2,3,4], because studies of this slamming phenomenon do not allow exact values to be obtained, but statistical values can be calculated. Polymer matrix composites with continuous glass fiber reinforcement (GFRP), which are widely used in the maritime sector because of their lightweight and suitable properties, can be seriously damaged by this type of impact.

To achieve a good energy absorption capacity, there are more strategies than selecting materials with high intrinsic resistance. The use of hybrid materials with complementary properties of the constituents, or the use of intricated geometries, allows much better results to be obtained. In this context, the use of new materials with novel architecture, such as auxetic structures with a negative Poisson ratio [5], which give them exceptional properties for the dissipation of impact energy, may represent a significant advance in the mitigation of this kind of problem. Many examples of authentic structures with re-entrant hexagonal geometry capable of absorbing energy with favorable results can be found in the literature. This structure is the most commonly used in macroscopic studies, and, in general, it shows the most favorable results in terms of the Poisson coefficient and energy absorption capacity [6]. This structure consists of symmetric concave hexagons that can give rise to geometries, both two-dimensional and three-dimensional, in different variants that simultaneously show auxetic behavior in several directions [7,8]. The double-V structure has also been successfully employed [7,9,10]. It consists of the concatenation of symmetrical quadrilaterals in an arrowhead. This type of auxetic material has a more favorable behavior under vertical loads than the reentrant hexagonal, in which the vertical elements tend to fail by buckling [11] in which Yan says, “auxetic materials or structures, especially 3D cellular lattice architectures with negative Poisson’s ratio (NPR) have attracted great attention due to their unprecedented mechanical behaviors and promising applications in recent years.” Auxetic behavior can also be achieved by connecting sets of equal polygons at their vertices by rotating joints [7], and Alderson et al. found that in chiral webs, based on rigid rings around which several tangent elements extend, auxetic behavior is generally more pronounced the more tangent elements start from each cylinder [12]. An evolution that was proposed for chiral auxetic materials is hierarchical chiral structures in which macroscopic auxetic metastructures are created using smaller chiral units. It was found that this type of second-order structure can significantly improve the shock absorption capacity of the material, and its properties can be more precisely tailored [13].

However, there are other possibilities for creating auxetic behaviors based on the principles of biomimicry and bioinspiration. Numerous examples of auxetic materials present in different situations in nature have been discovered in the last decades [14,15,16,17,18,19,20,21]. Moreover, these concepts are useful because many tough biological materials present a highly remarkable energy absorption capacity based on reinforcement mechanisms operating at different spatial scales in a hierarchical organization. An example of this is the prehistoric fish Polypterus senegalus, which is protected by its natural scale armor formed by four layers of reinforcement based on nanocomposites with gradient properties [22]. In addition, the study of the internal vascular structures of bamboo [23,24] or horsetail [25] has allowed the development of new energy dissipaters based on cylindrical concentric tubes connected in three by means of different bionic structures. These types of geometries are also present elsewhere in the natural world, such as in the elytra: mobile wings protecting the shells of some flying insects such as beetles [26,27]. Mother-of-pearl is also one of the biological protective structures that has aroused more interest over the years due to its excellent mechanical properties. With this material, some mollusks coat the inner side of their shells [28,29,30].

The interest in this category of materials, as impact energy dissipative layers, lies in the nature of their mechanical response to low-velocity (nonballistic) forces. Due to its configuration, the material in the zone affected by an impact significantly increases in density, thus increasing the value of its mechanical impedance. Hou et al. found that when they form part of laminates, auxetic materials exhibit higher structural integrity and, especially, higher durability than geometries with a positive Poisson’s ratio against low-velocity impacts [31]. Moreover, they are able to remain intact against a higher number of shock cycles. These results have been confirmed by other more recent studies [3,4], in which auxetic geometries have been found to have better mechanical properties under cyclic loading conditions, resist a greater number of cycles, have higher applied forces, and show a greater repeatability between cycles. Other numerical and experimental analyses have shown that auxetic materials also better resist impact penetration, increasing strength at low [6] and high velocities (for ballistic applications) [32]. In the latter study, the benefits of using these materials as reinforcements in carbon fiber laminates were assessed. It was also shown that creating composites with auxetic elements can result in metamaterials that exhibit general auxetic behavior while improving stiffness in certain directions [33].

For these reasons, this type of material is proposed as a solution to the slamming problem that occurs in high-speed boats, which sail and repeatedly impact against the water surface. In this paper, some dissipative layer models are presented, with the purpose of introducing them in fiberglass and polyester laminates, to partially dissipate the energy of impacts and reduce damage to the material that forms the hull of the vessel. To simulate the real in-service conditions of the boat, drop-weight impact tests were performed for different impact energies. When the damage induced in the material was evaluated, it was possible to characterize the behavior of the dissipative layers and determine the impact protection mechanisms and the range of energies in which they properly perform. The use of the impact dissipative layer was found to reduce damage in all the cases studied. With this study, an alternative of using auxetic layers to absorb the destructive damage of slamming was designed. These layers, once installed within GFRP laminates, will allow composite material to be protected from the damaging impacts that produce intralaminar micro-cracks and interlaminar delaminations. The use of auxetic layers in this type of area has not been considered before, and the demonstration of their benefits opens a new field of construction and design of GFRP structures.

## 2. Materials and Methods

For the development of the dissipative layer, different functional and design requirements were considered. For example, the lattice structure of the material must be able to withstand a high number of impact cycles at low velocity and must have a reduced weight and thickness to be able to form part of the laminated materials. In addition, its geometry must be compatible with the constraints of the fused deposition modeling (FDM) manufacturing technology.

The dissipative layer models developed, formed by the combination of two polymers with different stiffnesses, were gradually refined. Figure 1 shows the first proposal (Type A) consisting of thermoplastic polyurethane (TPU) as elastic material and polylactic acid (PLA) as rigid material. These materials were arranged so that the rigid material confined the elastic material, such that, upon impact, the π shape (rigid polymer) opened and confined the elastic material. TPU has a Poisson ratio close to 0.5, so it behaves as a quasi-incompressible solid. Unable to compress, TPU deforms upward toward the surface where the impact was received, trying to conserve volume. This mechanism redirects the energy of the impact, the elastic material acting as an energy container that absorbs the impact and dissipates the energy in the form of heat. In this way, the GFRP laminate underneath the dissipative layer is protected, reducing the generation of damage to the case, and extending the lifespan.

For this hybrid metastructure, with a rigid polymer with a π shape and a hyperelastic polymer that constituted the continuous phase, an auxetic material was obtained that mainly increased the bending stiffness of the assembly and returned the impact energy in the direction of the impacted surface. However, the main disadvantage of this dissipative layer is that it had a very large thickness, which is convenient to reduce because it was going to be introduced inside the GFRP laminate that forms the hull of a boat. Therefore, the design evolved into a second model of the dissipative layer (Type B), shown in Figure 2. With this configuration, the thickness was reduced, the PLA was replaced by nylon, which has more suitable mechanical properties to receive multiple impacts without degrading (better toughness), and the structure of the rigid polymer was modified by softening the angles to avoid stress concentrations that can break the element after a first impact. In this way, the structure of the metamaterial was improved, the dissipative layer was thinner, and the weight was reduced. This type was tested, and the results are presented in this paper.

For the GFRP laminate, 6 layers of 30 g/m^2^ type E fiberglass and polyester resin were used. The orientations were 0°/90° and +45°/−45°. Table 1 shows the characteristics of the laminate. Figure 3 shows the laminate with the impact dissipative layer located within the laminate.

The laminate was fabricated by vacuum resin infusion. For the Type B dissipative layer, additive manufacturing using dual-head FDM printing was used. To fix the layer to the laminate, a polyimide veil was placed on top of it, which did not play a structural role and was used only to hold the layer in position. In total, 10 × 10 cm laminates (10 samples) were made: 3 without a dissipative layer (reference laminates) and 7 with a dissipative layer. Figure 4 shows the GFRP laminate fabricated by vacuum resin infusion.

Drop-weight impact tests at different energies were performed according to ASTM D2444 and ISO 3127. Figure 5a shows the equipment used for impact tests. The equipment had an impact trolley that was dropped from a certain adjustable height onto the specimen and was connected to an electronic system to control the test. Figure 5b shows the impactor carriage, to which additional cylindrical weights attached to the impactor carriage could be added to obtain the required impact energy. This was calculated through the potential energy formula. The element that impacts the specimen was a half-inch-diameter steel sphere. The impactor was equipped with accelerometers whose recording allowed us to know the energy transmitted in the impact, the energy returned elastically, and the energy dissipated by the material after each test. The impact event occurred in milliseconds, so it was mandatory to work with a data acquisition system above 10 kHz. Tests were performed at five different impact energies: 35, 60, 70, 90, and 100 J. The selected energy values corresponded to the pressure peaks studied for the slamming phenomenon [1]. Considering the planing hull vessel for which this modification with auxetic layer is to be applied, when it hits the sea due to hydrodynamic lift and waves, it hits the bow with similar energy values, which we studied by pressure spectra and adjusted to these impact levels.

To assess damage after each impact, immersion ultrasonic inspection was used to measure the extent of the damaged area and to estimate the damage to the GFRP laminate. For this purpose, a TECNITEST (Madrid, Spain) automated system was used, which employed the pulse-echo ultrasonic technique and the C-Scan mode of representation. The machine head, which incorporated the ultrasonic emitter-receiver, had 3 degrees of freedom. The scanning system, where the sensor was connected to a SONATEST device, allowed the acquisition of data during inspection and a computer that generates the 2D image in real time in false color with different levels of ultrasonic attenuation in the impact zone, related to the damage generated in the impacted panel (C-Scan representation).

## 3. Results

Obtained from the test was the recorded acceleration versus time readings, which are plotted in Figure 6 for different impact energies (35, 60, 70, 70, 90, and 100 J) for GFRP laminates without a dissipative layer. Each specimen with the auxetic material was tested only once because the impact was measured according to the applied energy. The equipment was set to give a single impact. The measurement was made using a unidirectional accelerometer that measured the G forces of gravity at impact. Due to the previous preparation, all the tests were successful, and the equipment did not demonstrate any differences. In the tests, accelerometer readings were taken at a frequency of 10 kHz for 1.0 s starting 0.2 s from the onset of the drop. With this acceleration, the measurements were obtained with a time interval of t = 0.1 ms. In all cases, the acceleration readings were recorded as multiples of the acceleration of gravity, *g* = 9.81 m·s^−2^. It should be noted that Figure 6 does not show the maximum peak of the vertical impact at 60 J per scale, but it is indicated that the value did not exceed 130 on the acceleration scale.

From these curves, it is possible to obtain, by integration, the force versus displacement variation curve for each impact test. Figure 7 shows, as an example, the curves for the impact energies of 35, 60, and 100 J.

The upward curve in Figure 7, which is the part of the impact from the initial moment of contact to the maximum displacement reached by the panel, coincides with the highest point of the curve. Subsequently, there is a downward curve that is related to the rebound of the impactor and the return of the elastically stored energy in the material. The area between the two curves is the dissipated energy, which corresponds to the fraction that was transformed into damage of the material.

These absorbed and returned energies were plotted against time. Figure 8 shows the curves for four different energies, with and without the dissipative layer. The difference between the maximum energy transferred to the laminate (highest point of the curve) and the energy returned (plateau on the right of each curve) is the energy dissipated by the material, both with and without a protective coating.

At the end of each test, an initial visual inspection of the damage caused by the impactor was carried out, both on the impact side (front view) and on the rear face of the panel at different energy levels (Figure 9).

However, visual inspection was not sufficient, as it was necessary to quantify the damage. For this purpose, ultrasonic inspections of each panel were performed, allowing us to obtain an ultrasonic attenuation map to quantify the extent of the damage. In Figure 10, the C-scan before and after impact in each panel is plotted for the different impact energies. Comparison with the images in Figure 9 makes it evident that the damage occasionally extended beyond what is visible.

## 4. Discussion

From the results provided by the ultrasonic tests, the extent of the damaged area could be measured, and plots could be obtained for each impact energy. Figure 11 shows the percentage of the damaged area versus impact energy. The curve for panels protected with a dissipative layer is well below the curve for unprotected panels, which indicates that the layer effectively protected the panel.

In addition, the energy versus time was plotted to determine the energy absorbed by the material during impact and the energy returned (Figure 8). The difference, for each curve, between the maximum energy and the energy at the end of the test provides the energy dissipated by the material (with or without the dissipative layer) for each impact energy. Figure 12 shows the energy absorbed by the panels after impact versus the impact energy transmitted by the impactor head. Each point on the diagonal of this graph indicates that all of the energy was absorbed by the material. For points below the diagonal, the energy received was partially dissipated by the material (generating internal damage), and the remaining was elastically returned. The auxetic layer installed in the specimen shows that the level of energy absorbed from the modified specimen had a higher return of energy because it was not absorbed, as shown in Figure 12. The unmodified specimen was more damaged during the experimental part in the comparisons of the results. Therefore, the curve shows that it absorbed more energy. The images shown in Figure 9 corroborate these results.

Using the above information, we constructed curves that relate the energy returned by the material as a function of the impact energy received (Figure 13).

As seen in these graphs, the unprotected material partially returned (a maximum of 5%) the energy received in the impact, and only for very low impact energies of about 30 J. Above a certain impact energy, the material absorbed all the energy it received, converting it into internal damage. However, with the shielding layer, it was feasible to return a significantly higher percentage of the energy received during the impact (around 25%). In addition, the protection was effective up to higher impact energies (over 50 J). Future developments of these types of dissipative layers will involve the search for a more efficient management of the absorbed and returned energy during an impact event, looking to shift the curves upward (more energy is elastically returned) and to the right (protection efficiency up to higher energies). It is also important that the absorbed energy is diverted back to the surface, preventing damage to the GFRP composite plates, and causing damage only to the protective layer.

These results correspond to the images in Figure 10, in which the C-scan shows the evolution of the damage in the panel. The images shown in the range of colors show greater internal damage for unmodified panels than for modified panels. This allowed us to have a clearer understanding that the protection of the auxetic layer was effective for the layers of the laminate.

## 5. Conclusions

The use of impact energy dissipative layers is based on the combination of two polymers to form a metamaterial with auxetic behavior. One of the polymers has a higher stiffness (in this case, nylon) and confines another polymer, which is quasi-incompressible (TPU in the case presented). The use of the impact dissipative layer makes it possible to protect the GFRP laminate of the hull of a high-speed craft against impacts due to the slamming phenomenon. In all the cases studied, for different impact energies, damage to the structural material is reduced, extending its service life. The dissipative layer distributes the impact energy over a larger area, which reduces the energy per unit area reaching the laminate, avoiding serious damage such as microcracking of the polymer matrix, fiber breakage, and delamination. For impact energies up to 50 J, the protective coatings tested have shown to be effective, returning up to 25% of the energy received on impact. For energies above these values, there is piercing to the layer, and the efficiency of the energy dissipation processes is reduced. For impacts below 20 J, the shielding is total, and the laminate does not suffer internal damage.

The present study did not perform a quantitative evaluation but rather a qualitative one because a comparison of the benefits of using auxetic modification was conducted.

The present investigation demonstrates that a GFRP panel made with an auxetic layer will have a longer life. The level of protection compared with previous investigations [2] in which 2D viscoelastic sheets were used was improved. This new generation of protective layers using auxetic technology effectively protects planing hull vessel laminate from the slamming phenomenon.

## Figures and Tables

**Figure 1 polymers-14-03631-f001:**
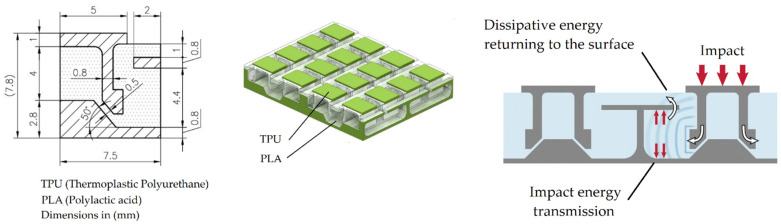
Type A model of the dissipative layer: PLA/TPU hybrid material with auxetic behavior.

**Figure 2 polymers-14-03631-f002:**
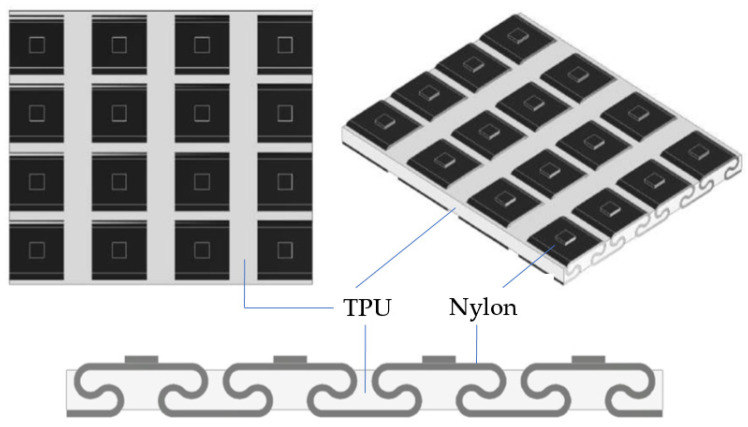
Model Type B of the dissipative layer: nylon/TPU hybrid metamaterial, partial S-hinge structure with auxetic behavior.

**Figure 3 polymers-14-03631-f003:**
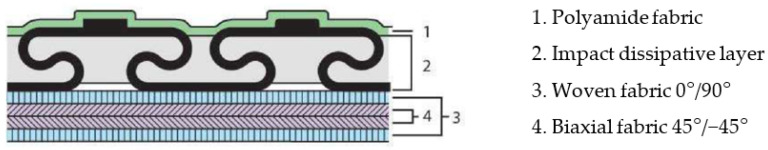
Stacking sequence and insertion of the dissipative layer of Type B.

**Figure 4 polymers-14-03631-f004:**
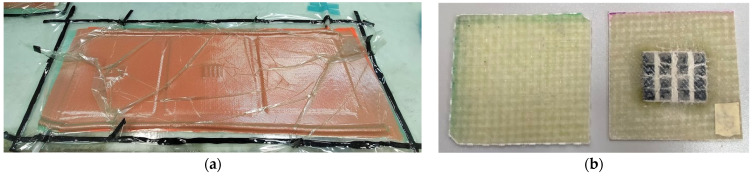
Fabrication of the laminate with dissipative layer: (**a**) GFRP laminate fabricated by vacuum resin infusion; (**b**) laminates without and with dissipative layer.

**Figure 5 polymers-14-03631-f005:**
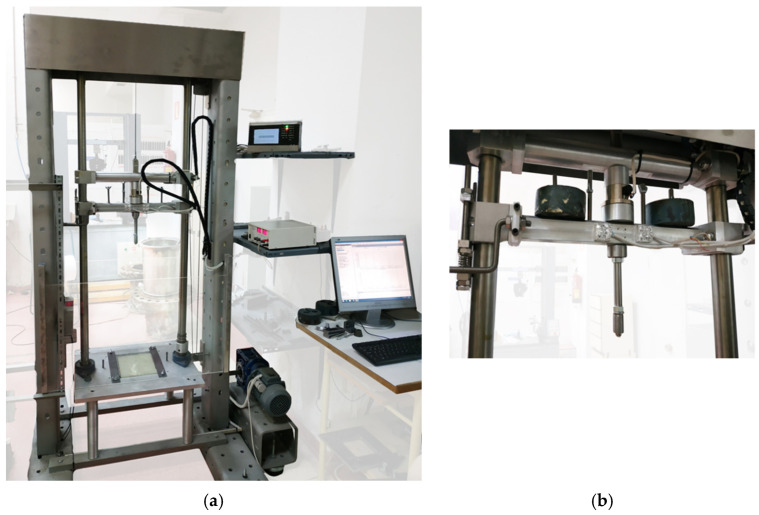
Equipment used for drop-weight impact tests: (**a**) frame and control; (**b**) impactor head.

**Figure 6 polymers-14-03631-f006:**
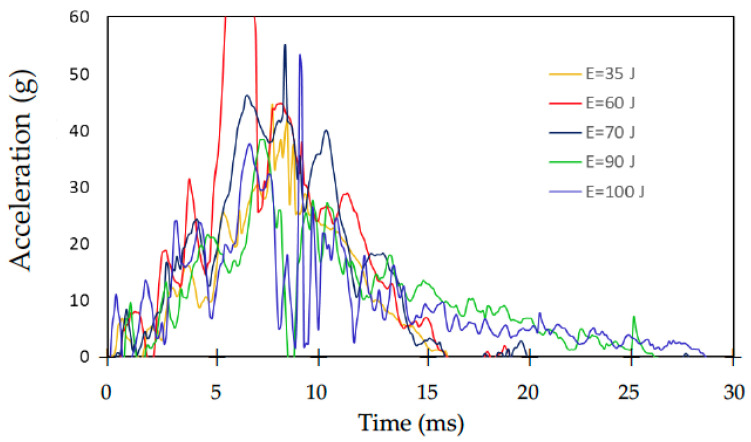
Acceleration versus time at different impact energies.

**Figure 7 polymers-14-03631-f007:**
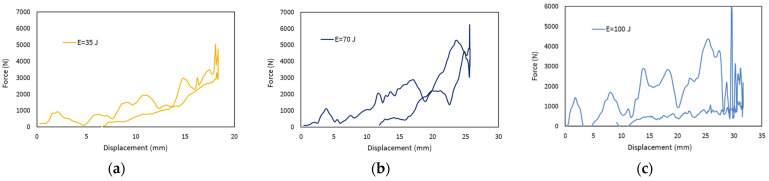
Force versus displacement curves for impact energies: (**a**) 35 J; (**b**) 70 J; (**c**) 100 J.

**Figure 8 polymers-14-03631-f008:**
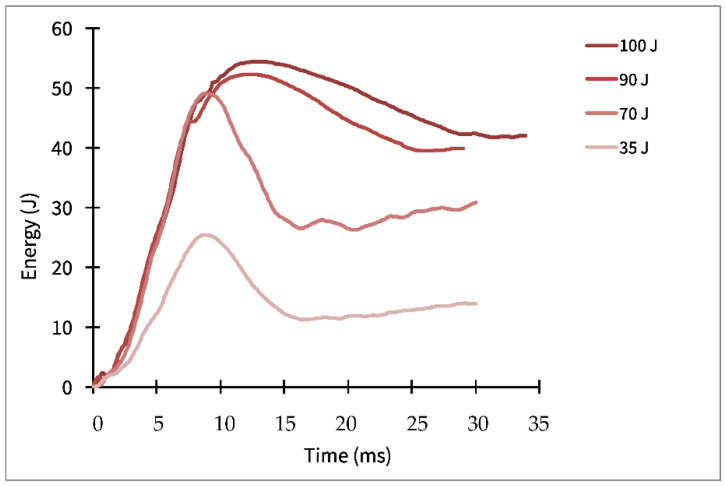
Energy dissipated versus time for three impact energies: 35, 70, 100 J.

**Figure 9 polymers-14-03631-f009:**
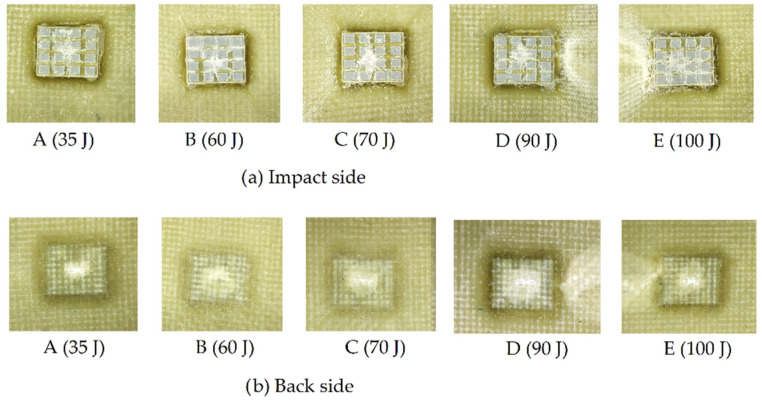
Visual inspection of damage on both sides.

**Figure 10 polymers-14-03631-f010:**
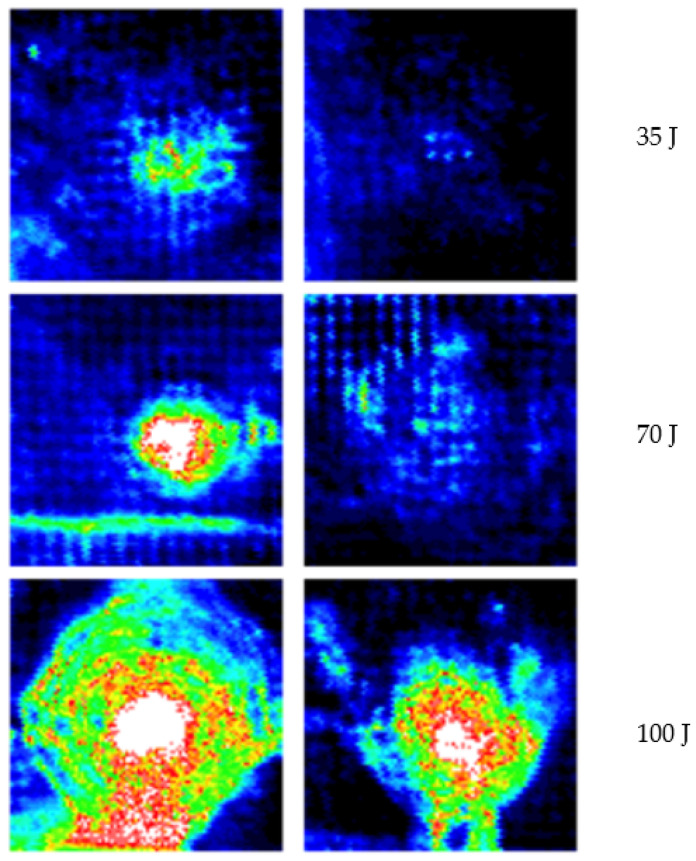
Ultrasonic inspection: C-scan maps of the ultrasonic attenuation after impact at 35 J, 70 J, and 100 J. Panels without protective layer (**left column**) and with protective layer (**right column**).

**Figure 11 polymers-14-03631-f011:**
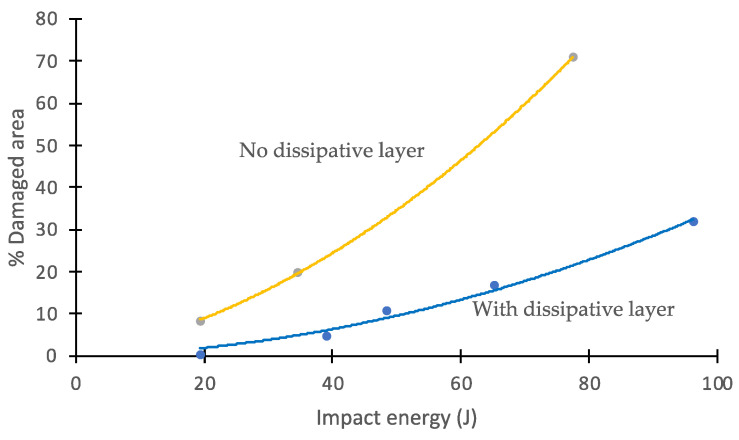
Percentage of damaged area vs. impact energy. Effect of the dissipative layer.

**Figure 12 polymers-14-03631-f012:**
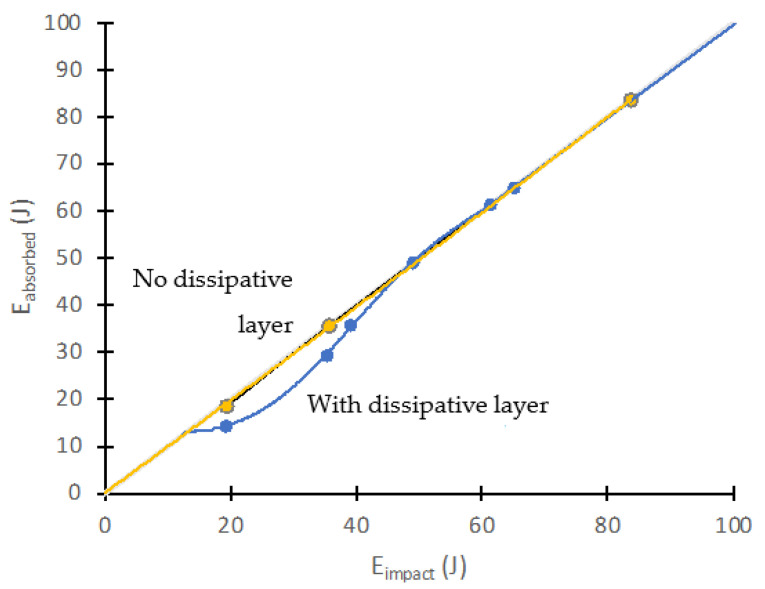
Received impact energy vs. absorbed energy. Shielding effect of the dissipative layer.

**Figure 13 polymers-14-03631-f013:**
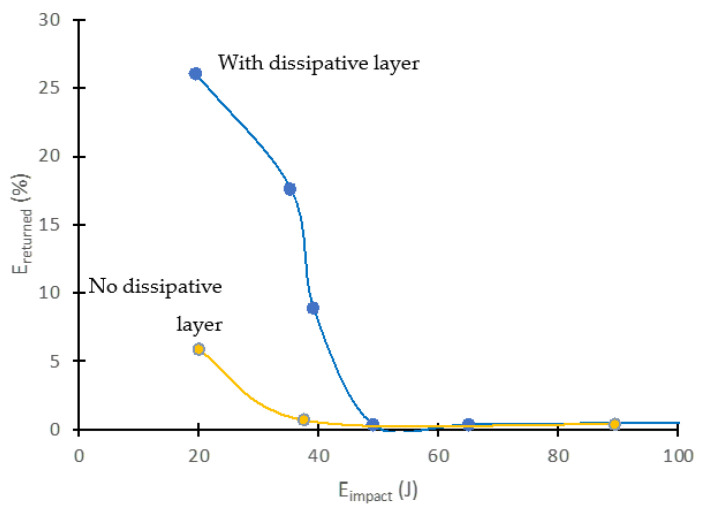
Impact peak energy vs. elastically returned energy. Shielding effect of the dissipative layer.

**Table 1 polymers-14-03631-t001:** Material properties of the hull laminate.

Material	Type
Surface Veil	Polyamide fabric
0°/90° twill fabric	UTR 581 T/100, 581 g/m^2^
+45°/−45° biaxial fabric	X450 E05C, 450 g/m^2^
Polyester resin	Crystic U904LVK30

## Data Availability

Not applicable.

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
