# Peer review of "Design, Fabrication, and Characterization of an Impact Dissipative Layer for Fiber-Reinforced Polymer Composites"

_polymers, 2022, doi:10.3390/polym14173631_

Round 1
Reviewer 1 Report
The authors presented an article « Design, Fabrication and Characterization of an Impact Dissipative Layer for Fiber-Reinforced Polymer Composites».
· Abstract
The abstract need to be improved. The abstract is written long. Shorter and core findings of the study should be given. Please provide the main quantitative and qualitative research core findings. Demonstrate in the abstract novelty, practical significance. Briefly list the input and output parameters of the research.
· Introduction
Seemingly, a comprehensive literature review was given. However, they were just summarized one- by-one. The authors have to stop after writing each example and think about the contributions and lack of knowledge for each paper. After that, in the final lines of the introduction give the blank spots of the topic. Then it will be clear what did authors make differently from the open literature.
In the last paragraph of the introduction section
What is the scientific novelty of the paper? What is the practical value? What makes this approach different from other researchers? Please specify. Gap and significance of the work must be included.
· 2. Materials and Methods
Please provide more detailed basis and reference for selecting impact energies. Please specify
What method of experiment planning is used and why?
Describe the measurement procedure in more detail. At what point in time? How is the measuring set up? How many times were the experiments repeated?
Why is this material so important?
Can't understand anything from Figure 5b.
· 3. Results
I think it's wrong page 6, line 194 “….(35, 60, 70, 70, 90 and 100J)”
In Figure 6, some data (E: 60J) did not fit into the graph.
It is useful to add explanations of parameters to the results obtained. At least five sentences for each Figures (Fig 7-9). The results obtained should be explained by supporting the literature.
Could you discuss the relationship between impact energy change and damage in more detail? What are phenomena? As a reference, please see:, https://doi.org/10.3390/polym14010095
· Conclusions
The conclusions need to be improved. The results are written too long. It is necessary to more clearly show the novelty of the article. Add qualitative and quantitative results of your work. What is the difference from previous work in this area? Show practical relevance. What are the differences from previous works?
· Authors should carefully study the comments and make improvements to the article step by step. All changes should be highlighted in color.
Author Response
Dear reviewer, I attach the following comments
- Abstract
The abstract need to be improved. The abstract is written long. Shorter and core findings of the study should be given. Please provide the main quantitative and qualitative research core findings. Demonstrate in the abstract novelty, practical significance. Briefly list the input and output parameters of the research.
The following paragraph was inserted:
The option of a very effective low-impact energy dissipating material applied to GFRP (Glass fiber reinforce plastic) composite laminates using auxectic technology is presented. In the case of planing hull vessels that use the same high-speed light materials that repeatedly impact the surface of the water when sailing, producing the slamming phenomenon. Research shows that the option to modify the laminate with auxectic layer protects the laminate from damage. This work proposes the manufacture of dissipative layers, introduced in laminates made with a polymeric matrix and fiberglass reinforcement, which are evaluated with weight drop tests under different impact energies. The collection and processing of data is done by a unidirectional gravitometer that gives the acceleration values of the impactor. With the tests, comparing unmodified panels with modified panels, it is shown that the energy absorbed by the unmodified panel is greater, at equal energy levels. The returned energy comparison curve is shown and the benefits of its use are presented
- Introduction
Seemingly, a comprehensive literature review was given. However, they were just summarized one- by-one. The authors have to stop after writing each example and think about the contributions and lack of knowledge for each paper. After that, in the final lines of the introduction give the blank spots of the topic. Then it will be clear what did authors make differently from the open literature.
The following paragraph was inserted:
phenomena that compromise the integrity of the materials of the vessels [1], in such a way that the author questions what is the next step now. The present investigation answers this question, because the planning hull vessels are the most selected ships for maritime use every day. One of these phenomena is slamming, which occurs when the bow of the
The following paragraph was inserted:
ship, and falling again when the hull collides with this fluid-free surface. This phenomenon produces, in the hull of high-speed boats, elevated pressure pulses of very short duration (on the order of milliseconds), localized on a very small surface as analyzed by researchers [2-4], because studies of this slamming phenomenon do not allow exact values to be obtained, but statistical values can be calculated. Polymer matrix composites with continuous glass fiber reinforcement (GFRP), which are widely used in the maritime sector because of their light weight and good properties, can be seriously damaged by this type of impact.
The following paragraph was inserted:
than the reentrant hexagonal, in which the vertical elements tend to fail by buckling [11] in which Yan says that “Auxetic materials or structures, especially 3D cellular lattice architectures with negative Poisson's ratio (NPR) have attracted great attention due to their unprecedented mechanical behaviors and promising applications in recent years.”.Auxetic behavior can also be achieved by connecting sets of equal polygons at their vertices by
In the last paragraph of the introduction section
What is the scientific novelty of the paper? What is the practical value? What makes this approach different from other researchers? Please specify. Gap and significance of the work must be included.
The following paragraph was inserted:
With this work, an alternative of using Auxetic layers to absorb the destructive damage of slamming is presented. These installed within GFRP laminates will allow the composite material to be protected from damaging impacts that produce intralaminar micro-cracks and interlaminar delaminations. The use of Auxetic layers in this type of uses has not been considered before and the demonstration of their benefits through research opens a new field of construction and design of GFRP structures.
- 2. Materials and Methods
Please provide more detailed basis and reference for selecting impact energies. Please specify
The following paragraph was inserted:
Figure 5b shows the impactor carriage, to which additional cylindrical weights attached to the impactor carriage can be added to obtain the required impact energy. This is calculated through the potential energy formul The selected energy values correspond to the pressure peaks studied for the slamming phenomenon [1]. The planing hull vessel for which this modification with auxetic layer is to be applied, when they hit the sea due to hydrodynamic lift and waves, they hit the bow with similar energy values which have been studied by pressure spectra and adjusted to these impact levels.
What method of experiment planning is used and why?
The following paragraph was inserted:
The selected energy values correspond to the pressure peaks studied for the slamming phenomenon [1]. The planing hull vessel for which this modification with auxetic layer is to be applied, when they hit the sea due to hydrodynamic lift and waves, they hit the bow with similar energy values which have been studied by pressure spectra and adjusted to these impact levels.
impact energies (35, 60, 70, 70, 90 and 100J) for GFRP laminates without a dissipative layer. Describe the measurement procedure in more detail. At what point in time? How is the measuring set up? How many times were the experiments repeated?
The following paragraph was inserted:
Each specimen with the auxetic material was tested only once since the impact is measured according to the applied energy. The equipment was set to give a single impact. The measurement was made using a unidirectional accelerometer that measures the G forces of gravity at impact. Due to the previous preparation, all the tests were successful and the equipment did not present any novelty
Why is this material so important? Can't understand anything from Figure 5b.
The following paragraph was inserted:
Figure 5b shows the impactor carriage, to which additional cylindrical weights attached to the impactor carriage can be added to obtain the required impact energy. This is calculated through the potential energy formula
- 3. Results
I think it's wrong page 6, line 194 “….(35, 60, 70, 70, 90 and 100J)”
This was corrected in the document
In Figure 6, some data (E: 60J) did not fit into the graph.
The following paragraph was inserted:
It should be noted that Figure 6 does not show the maximum peak of the vertical impact at 60J per scale, but it is indicated that its value does not exceed 130 on the acceleration scale.
It is useful to add explanations of parameters to the results obtained. At least five sentences for each Figures (Fig 7-9). The results obtained should be explained by supporting the literature.
The following paragraph was inserted:
The auxetic layer installed in the specimen shows that the level of energy absorbed from the modified specimen has a higher return of energy as seen in Figure 12, since it does not absorb it. The unmodified specimen was more damaged during the experimental part in the comparisons of results, therefore the curve shows that it absorbed more energy. The images shown in Figure 9 corroborate this information
Could you discuss the relationship between impact energy change and damage in more detail? What are phenomena? As a reference, please see:, https://doi.org/10.3390/polym14010095
The following paragraph was inserted:
These results agree with the images in figure 10, in which the C-scan shows the evolution of the damage in the panel. The images shown in the range of colors show greater internal damage for unmodified panels than for modified panels. This allows to have a clear idea that the protection of the auxetic layer is effective for the layers of the laminate protected by it.
- Conclusions
The conclusions need to be improved. The results are written too long. It is necessary to more clearly show the novelty of the article. Add qualitative and quantitative results of your work. What is the difference from previous work in this area? Show practical relevance. What are the differences from previous works?
The following paragraph was inserted:
The present study does not require a quantitative evaluation but rather a qualitative one, since a comparison of the benefits of using auxetic modification is presented.
The present investigation demonstrates that a GFRP panel made with an auxetic layer will have a longer useful life. The level of protection compared to previous investigations [2] in which 2D viscoelastic sheets were used, has been improved. This new generation of protective layers using auxetic technology effectively protects the planing hull vessel laminate from the slamming phenomenon.
- Authors should carefully study the comments and make improvements to the article step by step. All changes should be highlighted in color.
Reviewer 2 Report
The paper presents a very interesting research study. The developed materials behaviour during impact is promising for the mentioned application, but I think also for other domains. I do have a questions regarding the experimental study, or rather a recommendation. Considering that the materials are designed for boats hull applications, the authors should take into consideration the external polyamide layer behaviour in high humidity environments, considering that polyamide is absorbing water/ humidity from the external environment (even when the material is placed in a medium range humidity environment), that changes its mechanical properties and behaviour. The humidity conditions variations for impact resistance behaviour represent a factor that should be taken into consideration.
I think it is important to be mentioned in the study, at least that the factor was noted, or that future studies intend to analyze it (as humidity related issue did not seem to be the scope of the paper) but in order to be recommended specifically for this type of marine application, one can not overlook the water absorption issues.
Also, I suggest adding a small discussion specifically on the C-scan maps in Figure 10.
Author Response
Dear reviewer, I attach the following comments
The paper presents a very interesting research study. The developed materials behaviour during impact is promising for the mentioned application, but I think also for other domains. I do have a questions regarding the experimental study, or rather a recommendation. Considering that the materials are designed for boats hull applications, the authors should take into consideration the external polyamide layer behaviour in high humidity environments, considering that polyamide is absorbing water/ humidity from the external environment (even when the material is placed in a medium range humidity environment), that changes its mechanical properties and behaviour. The humidity conditions variations for impact resistance behaviour represent a factor that should be taken into consideration.
Humidity has not been considered for the investigation, because planning hull vessels are built in controlled environments. And the modification of the hull will depend on the areas and the location in which the layer is placed during the manufacturing process. I believe that it should not be mentioned in this article, but we will take it into consideration for future research.
I think it is important to be mentioned in the study, at least that the factor was noted, or that future studies intend to analyze it (as humidity related issue did not seem to be the scope of the paper) but in order to be recommended specifically for this type of marine application, one can not overlook the water absorption issues.
The following paragraph was inserted:
The present study does not require a quantitative evaluation but rather a qualitative one, since a comparison of the benefits of using auxetic modification is presented.
The present investigation demonstrates that a GFRP panel made with an auxetic layer will have a longer useful life. The level of protection compared to previous investigations [2] in which 2D viscoelastic sheets were used, has been improved. This new generation of protective layers using auxetic technology effectively protects the planing hull vessel laminate from the slamming phenomenon.
Also, I suggest adding a small discussion specifically on the C-scan maps in Figure 10.
The following paragraph was inserted:
These results agree with the images in figure 10, in which the C-scan shows the evolution of the damage in the panel. The images shown in the range of colors show greater internal damage for unmodified panels than for modified panels. This allows to have a clear idea that the protection of the auxetic layer is effective for the layers of the laminate protected by it.
